# Microsporidia infection alters *C. elegans* lipid levels

**Hala Tamim El Jarkass**[1][◉], **JiHae Jeon**[1][◉], **Nicholas O. Burton**[2], **Aaron W. Reinke**[iD][1]*

1 Department of Molecular Genetics, University of Toronto, Toronto, Canada, 2 Department of Metabolism and Nutritional Programming, Van Andel Research Institute, Grand Rapids, Michigan United States of America

◉ These authors contributed equally to this work.
* aaron.reinke@utoronto.ca

## Abstract

Microsporidia are fungal-related obligate intracellular parasites that infect many types of animals. Microsporidia have exceptionally reduced genomes resulting in limited metabolic capabilities and are thought to be reliant on host metabolism to fuel their own growth. Here, we investigate the impact of microsporidia infection on host lipid metabolism using the nematode *Caenorhabditis elegans* along with its natural microsporidian pathogen *Nematocida parisii*. We show that infection causes an increase in the level of *C. elegans* lipid droplet associated lipase, ATGL-1, and a decrease in host fat levels. A mutation that decreases ATGL-1 activity and over-expression of ATGL-1 did not significantly change *N. parisii* infection levels. Using lipidomics we show that *N. parisii* infection decreases *C. elegans* triglyceride levels and results in increased ceramides that we speculate are synthesized by *N. parisii*. Mutations in host genes involved in ceramide synthesis did not significantly change the levels of *N. parisii* infection. Together these results show that microsporidia can cause changes to lipid metabolism of their hosts, but some individual mutations of *C. elegans* lipid enzymes do not alter microsporidian growth.

## Introduction

Microsporidia are a large group of obligate intracellular parasites with the smallest known eukaryotic genomes [1–3]. Microsporidia infect most groups of animals and many species infect humans [4–7]. Microsporidia genomes are missing many metabolic enzymes, including those involved in lipid metabolism and some species have lost the ability to carry out glycolysis, making these parasites highly dependant upon their hosts for nutrients [8–10]. Microsporidia encode a number of transporters that allow them to acquire nucleotides and energy from their hosts [11–14]. Microsporidia also rely on other types of nutrients from their hosts, including lipids. Several lipids such as phosphatidic acid and linoleic acid were shown to be limiting factors

**Data availability statement:** All data needed to evaluate the conclusions in the paper are present in the paper and supplementary materials. Source Data are provided in S1 Data.

**Funding:** This work was supported by the Canadian Institutes of Health Research grant no. 400784 (to A. R.).

**Competing interests:** The authors have declared that no competing interests exist.

for microsporidia proliferation [15,16]. Microsporidia infection has a profound effect on metabolism in fruit flies, shrimp, crabs, and silkworms, causing a decrease in the levels of many types of lipids [15,17–20].

The free-living nematode *Caenorhabditis elegans* has been developed as a biological model system to study interactions with microsporidia [21]. Several species of microsporidia have been found to infect *C. elegans* either in the wild or in laboratory settings [10,22]. *Nematocida parisii* is the species most frequently found to infect *C. elegans* in the wild and is also the most commonly studied species in laboratory settings. [23]. This parasite has become a model system for microsporidia research and has been used to understand how hosts defend against infection [24–27], mechanisms of invasion [28], proliferation [16,29–31], spore exit [32], and to identify microsporidia inhibitors [16,33,34]. Infection begins when *N. parisii* spores are ingested by the worm, and the spore germinates using its unique infectious apparatus to deposit a sporoplasm inside of an intestinal cell [28,35]. The sporoplasms then divide and proliferate throughout the animal as a form called a meront [36,37]. During this process the 20 intestinal cells of *C. elegans* are fused together into a single cellular syncytium [29]. Sporulation then occurs, the spores exit without lysing the host plasma membrane, and the spores are defecated from the animal [32].

*C. elegans* fatty acid metabolism is regulated both by environmental stresses such as starvation and by bacteria. The majority of fat in *C. elegans* is stored as lipid droplets in intestinal epithelial cells [38]. Lipid droplets are involved in maintaining energy homeostasis and are important for longevity [39,40]. One of the major enzymes which regulates fat storage is the adipose triglyceride lipase, ATGL-1 [41]. Fasting causes an upregulation of ATGL-1 and a decrease in triglycerides [42,43]. Fatty acid synthesis is regulated in response to pathogen infection [44]. Several types of lipids, including cholesterol, and monosaturated and polyunsaturated fatty acids have been shown to modulate innate immunity in response to pathogens [45–47]. Additionally, pathogenic bacteria and yeast can cause a reduction in lipid droplets [44,48]. Bacteria that are part of the *C. elegans* microbiome can cause changes to fatty acid levels [37,49,50]. Additionally, bacterially produced sphingolipids have been shown to alter sphingolipid biosynthesis in *C. elegans*, providing protection against a pathogenic bacteria [51].

Here, we report the interactions between host lipid metabolism and microsporidia infection. We show that the lipase ATGL-1 is upregulated in response to *N. parisii* infection and that fat stores are depleted. Lipidomic profiling of infected animals reveals large changes in lipid composition including changes consistent with starvation. We also detect many novel ceramide species that appear to be made by *N. parisii*. As *N. parisii* has lost the enzymes to generate ceramides de novo, together these results suggest that *N. parisii* may be modifying host-provided lipids to synthesize these ceramides. We show that mutations in genes in the sphingolipid biosynthesis pathway did not change the amount of *N. parisii* infection. Overall, these experiments highlight changes to host lipid metabolism during microsporidia infection.

## Results

### ATGL-1 is upregulated and lipid stores are reduced in response to *N. parisii* infection

During fasting-induced nutritional deprivation of *C. elegans*, lipid stores are reduced [43]. This effect is largely mediated by the ATGL-1 lipase which colocalizes with intestinal lipid droplets and hydrolyzes triglycerides to form diglycerides [42]. To determine whether *N. parisii* induces a starvation-like state in *C. elegans*, we tested the effect of microsporidia infection on ATGL-1 expression using strain VS20, which contains an integrated transgenic array that overexpresses GFP-tagged ATGL-1 under its native promoter [43]. We infected the *atgl-1::gfp* strain with *N. parisii* for either 24, 48, or 72 hours. By 72 hours post infection (hpi), we observed a significant increase in ATGL-1 expression in infected animals (Fig 1A, B). To measure changes in the lipid stores, wild type (N2), *atgl-1::gfp*, and a reduction of function variant of this gene, *atgl-1* (*gk176565*) (ATGL-1 P87S*)*, were infected with *N. parisii* for 72 hours, fixed, and lipid droplets were stained using Oil Red O dye [43,52]. In wild-type animals there was a significant decrease in lipid staining upon *N. parisii* infection. We also observed a decrease in staining, though not significant, in the two *atgl-1* strains when they were infected with *N. parisii* (Fig 1C, D).

### ATGL-1 activity impacts the growth of *N. parisii*

To determine the role of ATGL-1 during infection with *N. parisii*, we analyzed the pathogen burden and the effect on the host in *atgl-1* mutant strains. We infected wild type, *atgl-1::gfp, and atgl-1* (*gk176565*) animals for 48 or 72 hours, then fixed each sample and stained with a Fluorescence In Situ Hybridization (FISH) probe specific for *N. parisii* 18S rRNA, which can label the intracellular meront stage [53]. We also stained with Direct Yellow 96 (DY96), which binds to the chitin of worm embryos and microsporidia spores [29,54]. We did not find any significant difference in the growth of *N. parsii* in either *atgl-1* strain at either time point (Fig 2A–C). To determine how infection impacts the host, we counted the number of embryos as a proxy for estimating the relative fitness of the worm (Fig 2D) [26,29]. There was a reduction in embryo number upon infection in all three strains, with infection causing a larger decrease in relative number of embryos in both ATGL-1 mutant strains, compared to N2 (Fig 2E).

### Lipidomics reveals impact of microsporidia on host lipid metabolism

To understand how *N. parisii* globally alters *C. elegans* lipid metabolism, we measured the lipidome of infected animals. Animals were grown in the presence or absence of *N. parisii* spores, these adults and their embryos were collected, and their lipids were measured using mass spectrometry. In total, 1451 different ionized lipids were reliably detected and 169 exhibited more than two-fold change with a significant P-value ($P < 0.01$) (Fig 3 and S1 Table). Infected animals displayed decreased levels of triglycerides and increased levels of diglycerides (Fig 3A). Long-chain acylcarnitines were also upregulated (Fig 3B). Phosphatidylcholines were most impacted by infection with some of these lipids exhibiting more than a 1000-fold change (Fig 3C). These highly upregulated phosphatidylcholines were not observed in another study of *C. elegans* lipids, and thus potentially represent metabolites generated by *N. parisii* [55]. Many d17 ceramides are downregulated in response to infection. In contrast, several d18 and d20 ceramides are highly upregulated (Fig 3D). These lipids are potentially synthesised by microsporidia as d17 is the only ceramide type known to be synthesized in *C. elegans* [56,57]. In embryos from infected and uninfected parents, 62 significantly regulated lipids were detected. In addition to the regulation of acylcarnitines, phosphatidylcholines, and ceramides, 14 phosphatidic acid molecules were upregulated in embryos from infected parents (S1 Fig). Overall, lipid profiling demonstrates that *N. parisii* infection causes a profound alternation to the lipidome of *C. elegans* in both adults and offspring.

### Mutations in *C. elegans* sphingolipid biosynthesis impact microsporidian infection

As we observed that putative novel ceramides may be synthesized by *N. parisii*, we sought to understand how host sphingolipid biosynthesis could impact *N. parisii* infection. We tested mutants for 9 out of 11 genes encoded by *C. elegans* which function

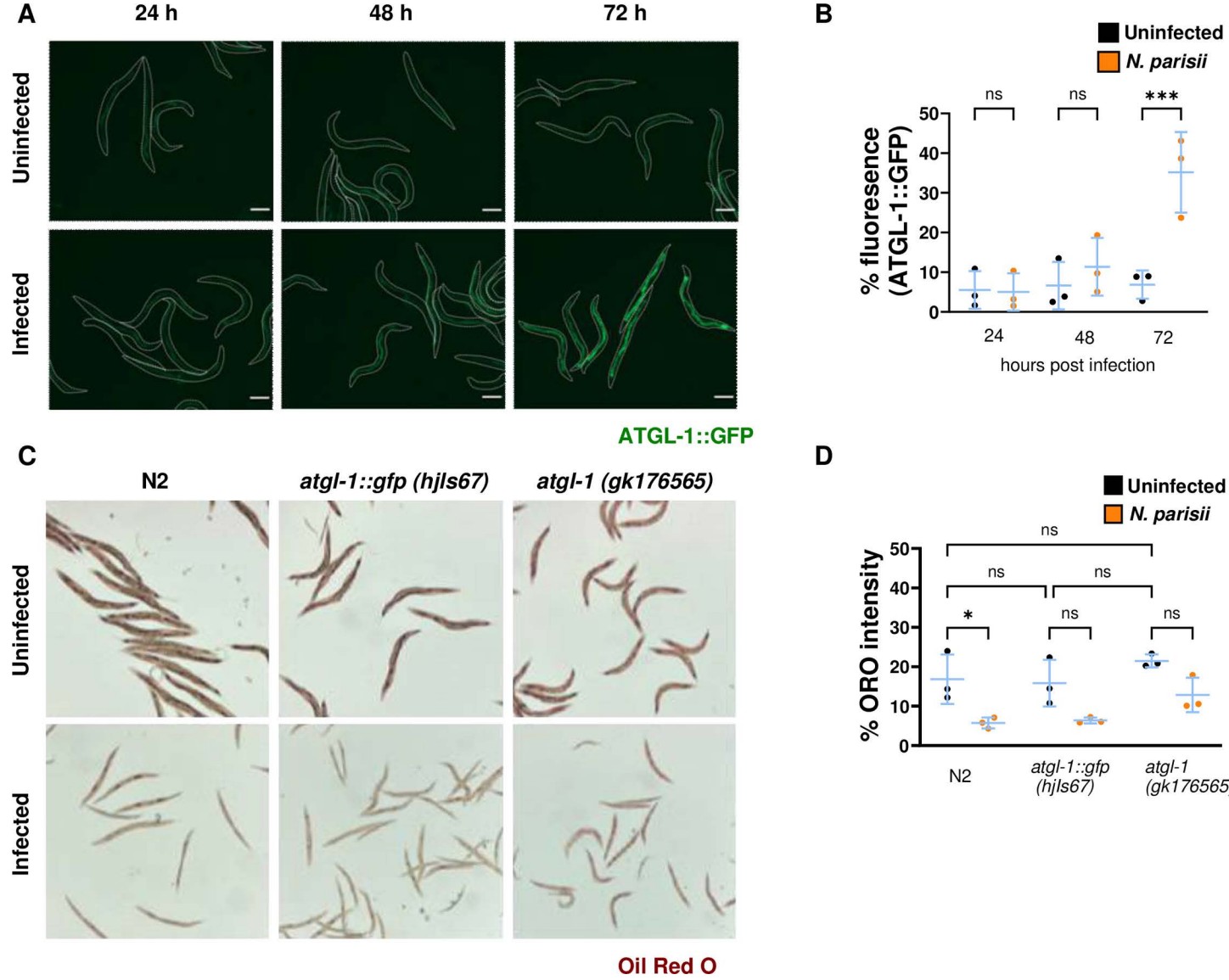

**Fig 1. ATGL-1 levels increase and lipid droplets decrease in response to *N. parisii* infection.** (A-B) *atgl-1::gfp* L1 stage *C. elegans* were cultured for a total of 72 hours and were either left uninfected or infected for 24, 48, or 72 hours. (A) Representative fluorescence images of live adult *atg-1::gfp* worms. White lines indicate the boundaries of worm bodies. Scale bars are 124 µm. (B) Quantification of ATGL-1::GFP. Data is from three independent replicates of 18−20 worms each. (C-D). L1 stage Wildtype (N2), *atgl-1::gfp, atgl-1 (gk176565)* worms of each strain were either infected or left uninfected for 72 hours and stained with Oil Red O. (C) Representative images of Oil Red O stained worms. (D) Quantification percentage Oil Red O intensity per worm. Data is from three independent replicates of at least 20 worms each. Mean±SD represented by horizontal bars. P-values were determined using one-way ANOVA with post hoc Šidák correction with tests between infected and uninfected samples of the same strain (A and B) and between uninfected samples of different strains (B). Significance is defined as * P<0.05, ** P<0.01, *** P<0.001, **** P<0.0001, and ns means not significant.

in the core sphingolipid biosynthetic pathway (Fig 4). Wild type and mutants deficient in sphingolipid biosynthesis were infected with *N. parisii* or left uninfected for 72 hours. We observed no significant changes in the levels of meronts or spores, except a small increase in the amount of spores in *asah-2* mutant animals (Fig 5A, B). To further probe if acid ceramidase mutants increase *N. parisii* infection levels, we tested both *asah-1* and *asah-2* for 48 hours and 72 hours. At the 48-hour time point we observed a slight increase in meronts in both mutants and we again observed a small increase in spores in *asah-2* at 72 hours,

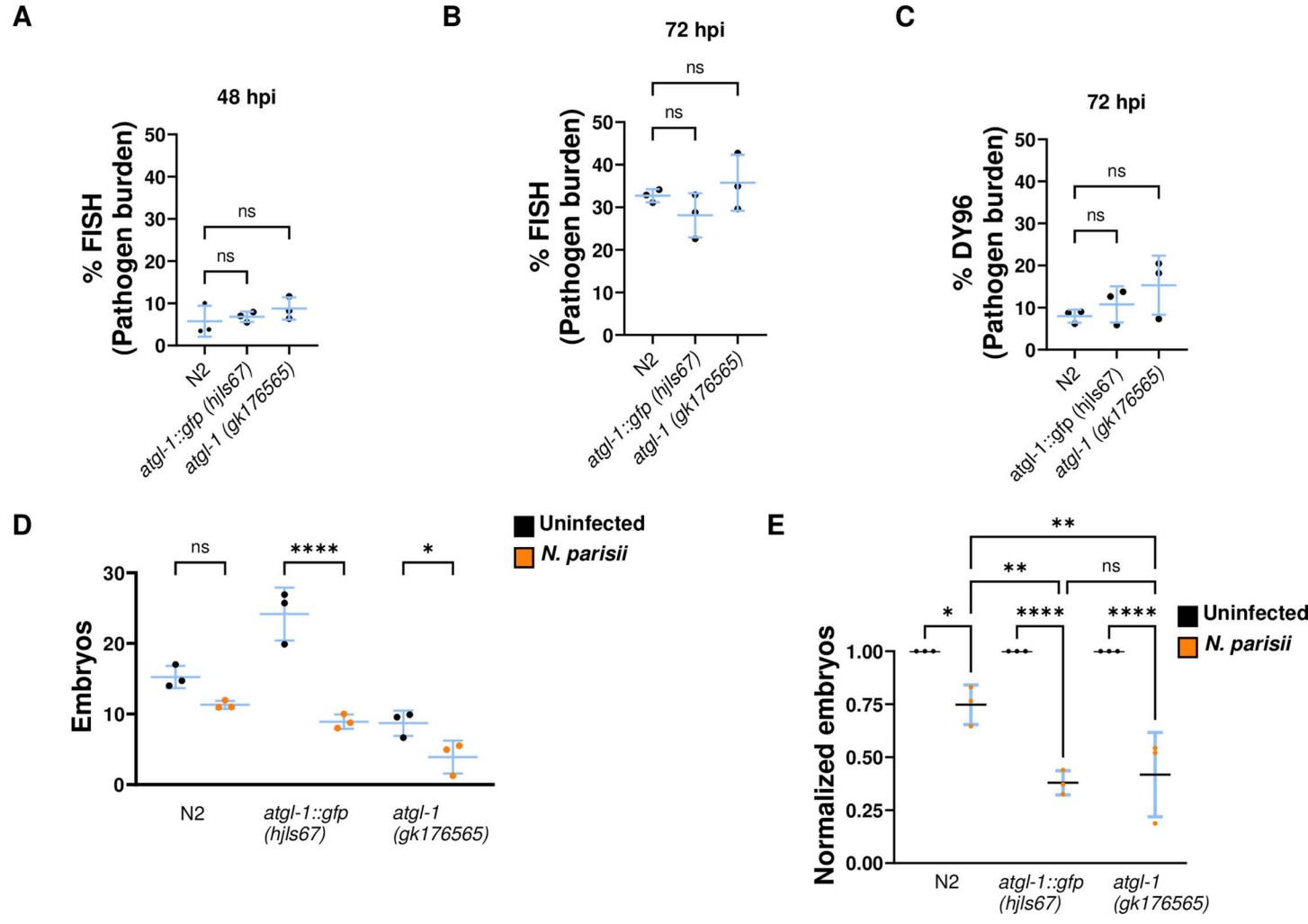

**Fig 2. Increased or reduced ATGL-1 activity does not significantly change *N. parisii* growth.** L1 stage wild-type (N2), *atgl-1::gfp, atgl-1 (gk176565)* worms were either not infected or infected with *N. parisii* for 48 **(A)** or 72 hours (D-E). Worms were then fixed and stained with a FISH probe against the *N. parisii* 18S RNA and DY96. (A-B) Quantitation of pathogen burden (meronts) per animal. (C) Quantification of pathogen burden (spores) per animal (D) Quantification of number of embryos per animal. (E) Number of embryos per worm, normalized to the uninfected condition for each strain. Data is from three independent replicates of at least 20 worms each. Mean±SD represented by horizontal bars. P-values were determined using one-way ANOVA with post hoc Dunnett's test with comparisons to N2 (A-C), Šidák correction (D and E) with tests between infected and uninfected samples of the same strain (D and E), and between infected samples of different strains (E). Significance is defined as * P<0.05, ** P<0.01, *** P<0.001, **** P<0.0001, and ns means not significant.

though none of these changes were statistically significant (Fig 5E, F). To determine if a reduction in sphingolipid biosynthesis enzymes affected the fitness of infected *C. elegans* we measured the number of embryos in sphingolipid mutants in the presence and absence of *N. parisii* spores (Fig 5C). We observed that the population of *ttm-5* mutant animals is significantly less gravid than N2, although these mutants have decreased levels of embryos in the absence of infection (Fig 5C, D).

## Sphingosine (d17:1) supplementation does not increase *N. parisii* proliferation

Ceramides are synthesised either through the desaturation of dihydroceramide or through ceramide synthase using sphingosine as a substrate (Fig 4). *N. parisii* encodes a ceramide synthase and could be generating ceramides by utilizing host

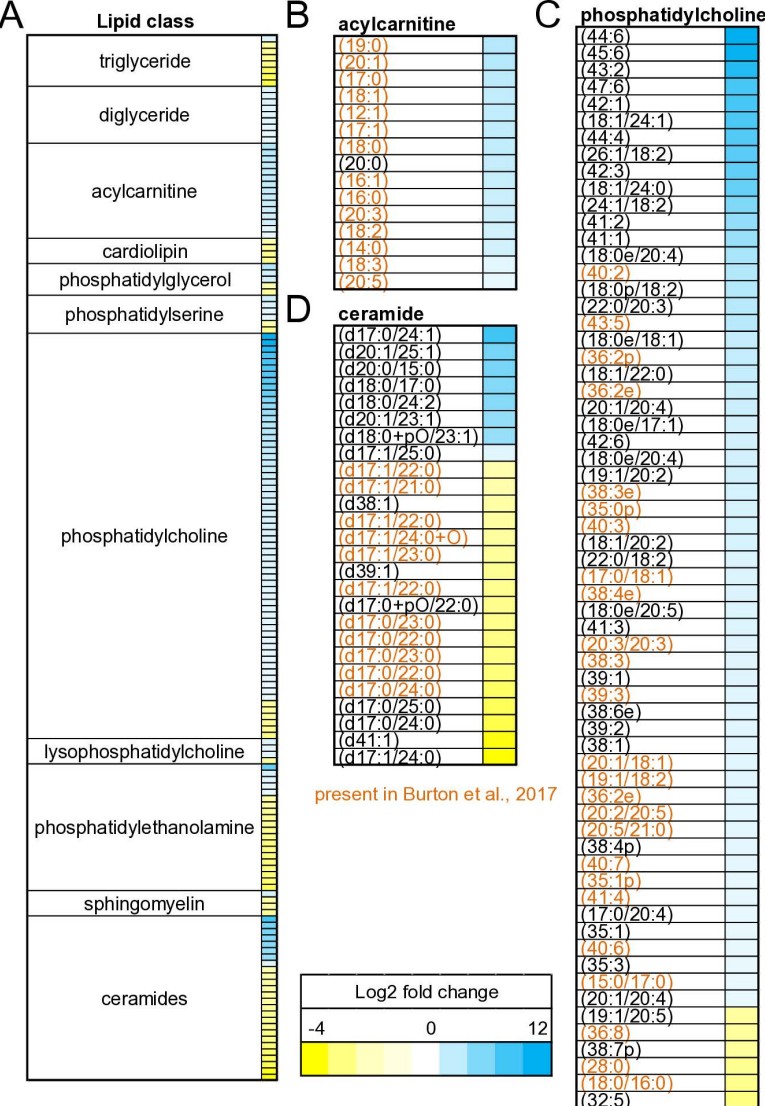

**Fig 3. *N. parisii* induces global changes to *C. elegans* lipidome.** (A-D) *C. elegans* were infected or left uninfected for 72 h and lipids measured using LC/MS. Heat map showing lipid metabolites that are upregulated (blue) or downregulated (yellow) upon infection at least 2-fold with a significant P-value determined by Student's T-test (P<0.01). **(A)** All classes of lipids with 2 or more lipids significantly up or down regulated. **(B)** Acylcarnitines. **(C)** Phosphatidylcholines. **(D)** Ceramides. Those lipids detected in Burton et al. 2017 colored in orange [55].

sphingosine. To test if exogenously added sphingosine could play a role during microsporidian infection, we performed infections using media supplemented with sphingosine. Wild-type *C. elegans* were exposed to *N. parisii* for 72 hours on 50 µM sphingosine or control plates containing ethanol. We observed no increase in the amount of spores generated when infected animals were supplemented with sphingosine (Fig 5G).

## Discussion

To understand if lipids are involved during *N. parisii* infection of *C. elegans*, we determined how infection alters lipid biosynthesis. We show that *N. parisii* infection upregulates ATGL-1, decreases the levels of some lipids, including

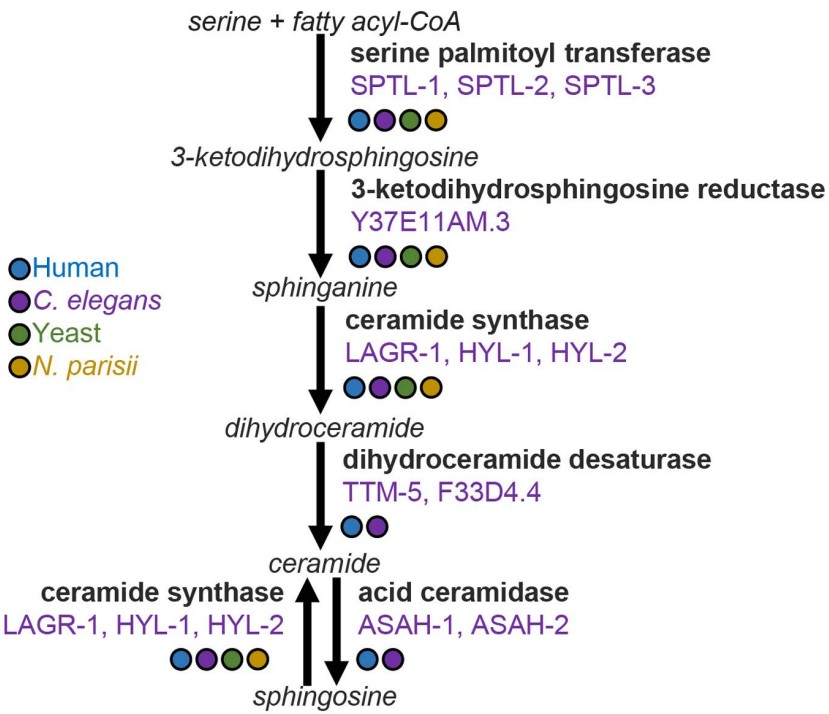

**Fig 4. Sphingolipid biosynthetic pathway.** Schematic of sphingolipid biosynthesis pathway. Metabolites are indicated in italics, enzymes indicated in bold, and *C. elegans* enzyme names are indicated in purple [58]. Presence of enzymes from humans, *C. elegans*, yeast (*Saccharomyces cerevisiae*), and *N. parisii* was determined with BLAST, using a threshold E-value of greater than $10^{-5}$ [59]. The presence of the enzyme is indicated by a colored circle corresponding to the legend at the left.

triglycerides, and increases the levels of many types of lipids, including diglycerides, phosphatidylcholines, and acylcarnitines. We also observed that modifying ATGL-1 activity, reducing the function of some sphingolipid biosynthesis genes, or supplementing with sphingosine did not result in significant changes to *N. parisii* growth. Altogether these results suggest that lipid metabolism is affected by infection, but changes to some lipid levels does not have a large impact on microsporidia growth. One limitation of our results is that there are several steps of sphingolipid biosynthesis where *C. elegans* encodes paralogous enzymes, which could lead to redundancy that prevented us from seeing an effect on microsporidia infection. Additionally, we observed that sphingosine supplementation did not increase *N. parisii* growth and it is possible that using a higher concentration, or using the branched form of sphingosine which is synthesized by *C. elegans* could increase microsporidia proliferation [56].

Large changes to lipid metabolism occur during microsporidia infection. There are several effects of *N. parisii* infection that are similar to starvation. When exposed to fasting conditions, *C. elegans* display upregulated ATGL-1, decreased Oil Red O staining, and decreased triglyceride levels, which are all changes we observed upon *N. parisii* infection [42,43]. We also observed acylcarnitines being upregulated during *N. parisii* infection, which has been shown to occur during starvation [60]. However, there are also differences, such as cardiolipin upregulation during starvation and we observed a downregulation of cardiolipins upon *N. parisii* infection [60]. Infection of *C. elegans* with *N. parisii* and other microsporidia did not show upregulation of *atgl-1* mRNA [61–63]. In fasted worms, *atgl-1* has been proposed to be post-transcriptionally regulated as there is a larger increase in protein than mRNA, and a similar regulation may be occurring during *N. parisii* infection [42]. We observed that some lipid mutants have a decrease in embryo production due to microsporidia infection. We previously showed that vitamin B12 provides tolerance to *N. parisii* infection and this suggests that depletion of other

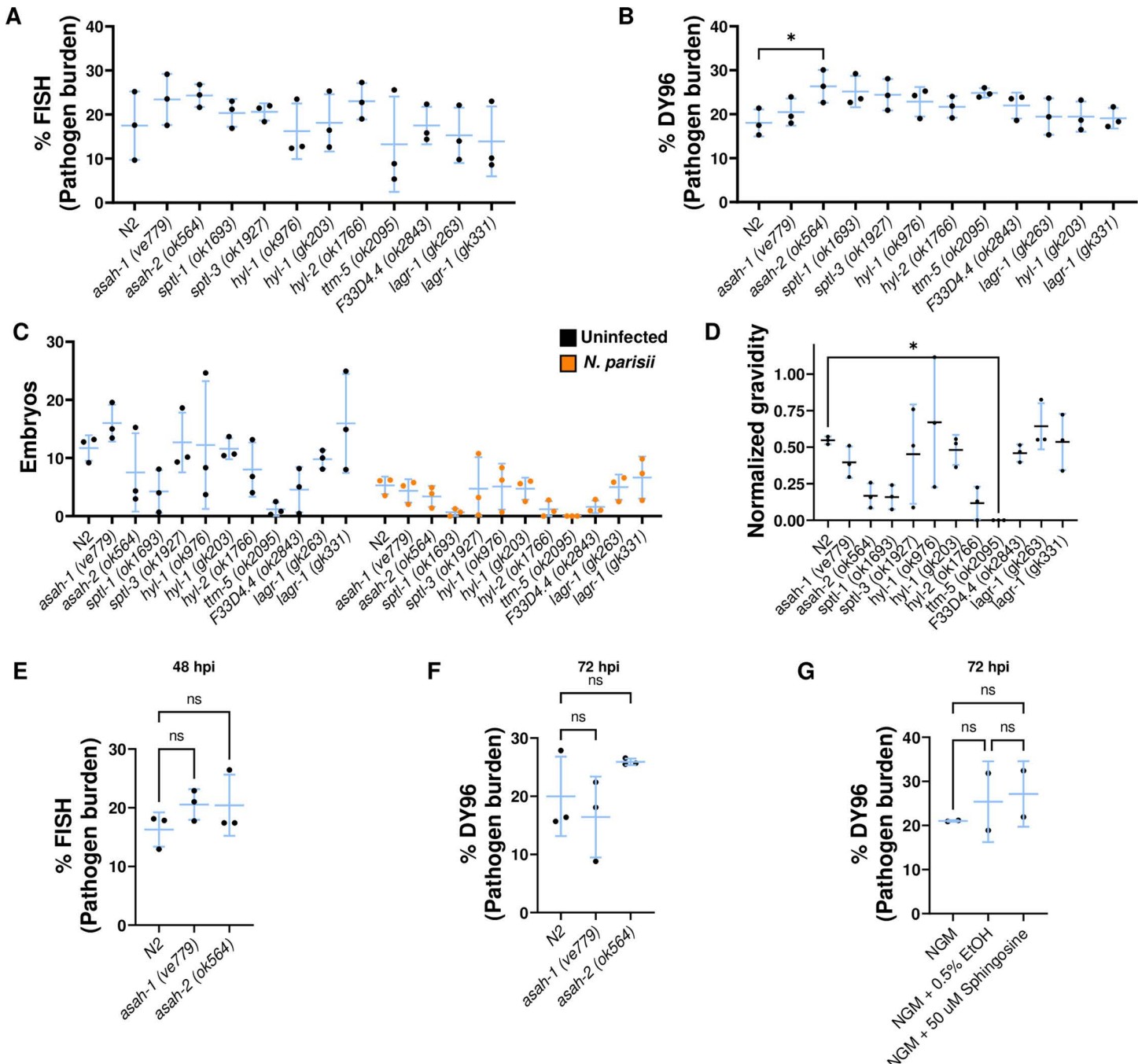

**Fig 5. *N. parisii* infection is not significantly changed in *C. elegans* sphingolipid biosynthesis mutants.** (A-D) L1 stage N2 or indicated mutant worms were either not infected or infected with *N. parisii* for 72 hours. Worms were then fixed and stained with a FISH probe against the *N. parisii* 18S and DY96. (A) Quantitation of pathogen burden (meronts) per animal. (B) Quantification of pathogen burden (spores) per animal. (C) Quantification of number of embryos per animal. (D) Quantification of population gravidity (gravidity is defined as an animal carrying at least 1 embryo), normalized to the uninfected condition for each strain. (E-F) L1 stage wild-type (N2), *asah-1 (ve779)*, *asah-2 (ok564)* worms were infected with *N. parisii* for 48 (E) or 72 hours (F), fixed, and stained with FISH probe and DY96. (E) Quantitation of pathogen burden (meronts) per animal. (F) Quantification of pathogen burden (spores) per animal. (G) L1 stage N2 animals were infected on control or sphingosine containing medium for 72 hours, fixed, and stained with DY96. Pathogen burden (spores) was quantified. Data is from three independent replicates of at least 20 worms each, except for (G) which had two replicates. Mean±SD represented by horizontal bars. p-values were determined using one-way ANOVA with post hoc Dunnett's test with comparisons to N2 and those that have a p>.05 and are not shown (A-D), Šidák correction with tests between infected and uninfected samples of the same strain (E), and Tukey's multiple comparison between all samples (F). Significance is defined as * P<0.05, ** P<0.01, *** P<0.001, **** P<0.0001, and ns means not significant.

nutrients can alter the impact of microsporidia infection on the host [64]. Infection with microsporidia also induces an inter-generational response which protects progeny from microsporidia infection [26]. The mechanisms of this intergenerational response are mostly unknown, and it is possible that the changes we see in lipid levels in offspring from infected animals may play a role in this immunity. Additionally, this work suggests that the addition of fatty acids maybe useful to enable continuous cell culture of some microsporidia, such as *E. bieneusi*, where so far this has not been possible [65]. *Cryptosporidium parvum* has been reported to be able to grow in a cell line that has higher levels of fatty acids and *Plasmodium knowlesi* growth is promoted by stearic acid [66,67]. Proliferation of microsporidia In *C. elegans*, fruit flies, and silkworms has been enhanced by addition of different lipids, and this same strategy may be useful with *E. bieneusi* [15,16,20]. The strong impact of microsporidia on lipid metabolism we observe in *C. elegans*, is also present in many other microsporidia hosts, suggesting that this is a common and conserved feature of microsporidia infection [17–20,68].

Sphingolipids are an important class of lipids that are components of membranes and used as signaling molecules [69]. Ceramides are a central type of sphingolipid that is made up of a sphingoid base and a fatty acyl chain [70]. *C. elegans* generate 17 carbon ceramides (d:17:1) and dihydroceramides (d:17:0) which contain a branched structure [56]. In several studies profiling lipids in *C. elegans*, d:17:1 and d:17:0 ceramides with different length of fatty acyl chains have been reported, but to the best of our knowledge, other sphingoid base branch lengths such as d:18 and d:20 have not been observed [55,71–73]. Upon infection we detected a downregulation of many d:17 ceramides species and upregulation ~30–80 fold of 6 lipid species with either d:18 or d:20 sphingoid bases. Although we have no direct evidence, we speculate that the most likely explanation for this result is that these longer sphingoid bases are generated by *N. parisii* enzymes. Other species of fungi have been observed to generate ceramides containing d:18 and d:20 sphingoid bases [74,75]. Most microsporidia, like *N. parisii*, encode a partial ceramide synthesis pathway, but do not have the ability to make ceramides de novo, whereas *Edhazardia aedis* contains the full ceramide synthesis pathway and *Enterocytozoon bieneusi* has lost the entire pathway.

In conclusion, we provide evidence that *C. elegans* lipid biosynthesis is affected by *N. parisii* infection. We have previously shown that mutants in the fatty acid desaturase *fat-2*, which is responsible for synthesising linoleic acid, delays *N. parisii* infection [16]. Future work using other lipid mutants or using a combination of mutants and RNAi for steps with multiple enzymes is likely to uncover other pathways necessary for *N. parisii* growth. Additionally, labeling *C. elegans* lipids could be used to determine if and how *N. parisii* can modify host lipids to generate ceramides [76].

## Materials and Methods

### Nematode strains and microsporidia culture

All *C. elegans* strains were maintained at 21 °C on Nematode Growth Medium (NGM) plates seeded with *Escherichia coli* OP50–1, as previously described [77]. All strains were derived from the Bristol N2 wild-type animals and obtained from the Caenorhabditis Genetics Centre (CGC). Details of strains used are listed in S2 Table. Synchronized populations of worms were generated by treating mixed stage populations with NaOH and hypochlorite, washing with M9, and hatching with rotation for 18–24 hours at 21 °C. *Escherichia coli* OP50–1 was grown at 37 °C for 18–24 in LB and then concentrated to make 10x stocks. *N. parisii* (ERTm1) spores were prepared as previously described [26]. Briefly, infected populations of *C. elegans* were harvested, disrupted with 2 mm diameter zirconia beads, and filtered (5 µm). Spores were confirmed to be free of bacterial contamination, counted by staining with DY96 and using a sperm counting slide, and frozen in aliquots at −80 °C.

### Infection and fluorescence imaging of ATGL-1::GFP

L1 stage *atgl-1::gfp* worms were divided into three groups and infected or left uninfected for 24, 48, or 72 h. For the 72 h infected group, 1,000 L1 animals were mixed with 1 million spores and 250 µl 10x OP50 and placed onto 3.5-cm NGM

plate. For the 24 h and 48 h infected groups, 1,000 L1 stage worms were first grown on NGM plates seeded with OP50−1 for 48 h and 24 h, respectively. The worms were then washed in M9 and mixed with 1 million spores and 250 µl 10X OP50 and placed onto 3.5-cm NGM plates. Uninfected controls were treated the same way as the infected populations except no *N. parisii* spores were added. Plates were incubated at 21 °C for a total of 72 hours. Worms were then paralyzed in 1 mM levamisole for 2 minutes and then transferred onto a freshly prepared 3% agarose pad on glass slide. Imaging was performed using a ZEISS Axio Imager 2 and the level of GFP was quantified for each animal and normalized by worm size (net fluorescence divided by its total area) using FIJI 2.1.0 [78].

### Oil red O staining and imaging

1,000 L1 stage animals, 400 µl of 10x *E. coli* OP50 and 3.5 million spores of *N. parisii* were mixed, plated onto a 6-cm unseeded NGM plate and incubated at 21 °C for 72 hours. Uninfected controls contained a volume of M9 equivalent to that used for spores. After 72 hours, animals were washed with PBST, fixed with 40% isopropanol stained with Oil Red O solution as previously described [79]. Animals were mounted onto glass slides and imaged using a ZEISS Axio Imager 2 with a mobile phone camera to capture stained animals. Oil red O levels were quantified using FIJI by first drawing around individual animals to designate regions of interest, followed by thresholding and measuring stain intensity.

### *N. parisii* infection of lipid mutants

L1 stage animals were mixed with OP50 and *N. parisii* spores and placed onto an NGM plates. Infections of synchronized *atgl-1::gfp* and *atgl-1 (gk176565)* worms were done with 1,000 L1s, 1.5 million spores, and 250 µl 10x OP50 on a 3.5-cm plate. Infection of all other lipid mutants was done with 1,000 L1s, 3.5 million spores, and 400 µl 10x OP50 on a 6-cm NGM plate. Infection plates were incubated at 21 °C for either 48 or 72 hours, depending on the experiment. Worms were washed off plates with M9 + 0.1% Tween-20 and fixed with acetone.

### DY96 and FISH staining

To stain spores and embryos, fixed worms were washed twice with 1 ml PBS + 0.1% Tween-20 then incubated in 500 µl of DY96 staining solution (1x PBS, 0.1% Tween-20, 20 µg/ml DY96, and 0.1% SDS) for 30 minutes at 21 °C. To stain microsporidia meronts, fixed worms were washed twice with 1 ml PBS + 0.1% Tween-20 and once with 1 ml hybridization buffer (900 mM NaCl, 20 mM Tris [pH 8.0], 0.01% SDS in dH2O). Supernatant was then removed, 100 µl hybridization buffer containing 5 ng/µl FISH probe was added and the worms were incubated at about 46 °C overnight. The probe used in the study was MicroB with Cal Fluor Red 610 attached (CTCTCGGCACTCCTTCCTG) [53]. Samples were washed once with wash buffer (hybridization buffer containing 5 mM EDTA and 20 µg/mL DY96). Worms were incubated in 500 µl wash buffer at 46 °C for 30–60 minutes. Supernatant was removed and stained worms were mounted on glass slides using 20 µl Everbrite containing DAPI for imaging with a ZEISS Axio Imager 2. Spore and meront fluorescence was quantified using FIJI 2.1.0 by thresholding fluorescence intensity and the remaining fluorescence signal was measured as percentage of the nematode body area [54,78].

### Lipidomics of *N. parisii* infected *C. elegans*

Synchronized L1 N2 animals were prepared by sodium hypochlorite treatment of mixed populations of worms. Infections were carried out by adding 10,000 worms, 5 million *N. parisii* spores, and 1 ml 10x OP50−1 per plate onto 13 10-cm NGM plates. For uninfected animals, 2,500 animals and 1 ml 10x OP50−1 were added per plate onto 10 10-cm NGM plates. Animals were incubated at 21°C for 72 hours. For adult samples, worms from 1 plate of each condition were washed into 1.5 ml tubes and then washed twice with 1 ml M9. To remove embryos and L1s, worms were allowed to settle for 1 minute, supernatant was removed, and 1 ml M9 was added. This was done a total of 4 times. Samples were then frozen in liquid nitrogen.

 

To prepare embryos, the remaining infected and uninfected plates were treated with sodium hypochlorite in 15 mL conical tubes and washed 3 times with 10 ml M9 and resuspended in 5 ml M9. To remove carcasses, tubes were left to settle for 4 minutes, supernatant removed and washed twice with 10 mL M9. Embryos were then added to a 10-cm NGM plate and incubated for 3 hours at room temperature. Embryos were washed off plates with M9 into 1.5 ml tubes and froze in liquid nitrogen. Experiment was performed in triplicate.

Liquid chromatography separation with mass spectrometry detection (LC–MS) of intact lipid species was performed as described previously [80].

### Sphingosine supplementation

Sphingosine (d17:1, from Cayman Chemicals) was dissolved in 100% ethanol as a vehicle to form a stock of 10 mM. This stock was then used to supplement sphingosine to NGM plates post autoclave to a final concentration of 50 uM sphingosine and 0.5% ethanol. This concentration of sphingosine was chosen based on a previous study performing sphingosine complementation in *C. elegans* [56]. Infections were performed using 400 L1 stage N2 animals, 160 µl of 10x OP50, and 1 million *N. parisii* spores on a 3.5-cm plate. After a 72-hour incubation at 21 °C, worms were harvested, stained with DY96, and imaged as described above.

### Statistical analysis

All data analysis was performed in GraphPad prism 10.4.1. Normality of data from three biological replicates was determined using Shapiro–Wilk test, and all data was normal with a P-value of $> 0.025$, except for lagr-1 (gk263) in Fig 5D and asah-2 in Fig 5E. P-values were determined using one-way ANOVA and using either post hoc Dunnett's, Šidák, or Tukey's test with significance defined as $< 0.05$. The one exception is the lipidomic data where significance was determined with Student's t-test and defined as $< 0.01$. The test used for each comparison is included in the figure legends.

### Supporting information

**S1 Fig.  *N. parisii* infection of *C. elegans* parents alters lipids in offspring.**
(PDF)

**S1 Table.  Lipidomic data from infected and uninfected adults and their embryos.**
(XLSX)

**S2 Table.  List of *C. elegans* strains used in the study.**
(PDF)

**S1 Data.  All infection data from the study.**
(XLSX)

### Acknowledgments

We are grateful to Meng Xiao, Yin Chen Wan, and Jonathan Tersigni for providing helpful comments on the manuscript. *C. elegans* strains were provided by the *Caenorhabditis* Genetics Center and we thank WormBase.

### Author contributions

**Conceptualization:** Nicholas O Burton, Aaron W. Reinke.

**Funding acquisition:** Aaron W. Reinke.

**Investigation:** Hala Tamim El Jarkass, JiHae Jeon, Nicholas O. Burton, Aaron W. Reinke.

**Visualization:** Hala Tamim El Jarkass.

**Writing – original draft:** JiHae Jeon, Aaron W. Reinke.

**Writing – review & editing:** Hala Tamim El Jarkass, JiHae Jeon, Nicholas O. Burton, Aaron W. Reinke.

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
