## [Decision Letter · Decision Letter 0]

Dear Dr. Reinke,

Thank you for submitting your manuscript to PLOS ONE. After careful consideration, we feel that it has merit but does not fully meet PLOS ONE’s publication criteria as it currently stands. Therefore, we invite you to submit a revised version of the manuscript that addresses the points raised during the review process.

**The reviewers provided positive reviews about the work and only require minor revision. I don't feel additional experimental work is needed. ****Please address the reviewers' points and submit your revised version. **  

We look forward to receiving your revised manuscript.

Kind regards,

Paul Dean

Academic Editor

PLOS ONE

2. Please update your submission to use the PLOS LaTeX template. The template and more information on our requirements for LaTeX submissions can be found at http://journals.plos.org/plosone/s/latex

“This work was supported by the Canadian Institutes of Health Research grant no. 400784 (to A. R.)”

“We are grateful to Meng Xiao, Yin Chen Wan, and Jonathan Tersigni for providing helpful comments on the manuscript. C. elegans strains were provided by the Caenorhabditis Genetics Center, which is funded by the National Institutes of Health (NIH) Office of Research Infrastructure Programs Grant P40 OD010440 and we thank WormBase. Funding: This work was supported by the Canadian Institutes of Health Research grant no. 400784 (to A. R.) Competing interests: The authors declare that they have no competing interests. Data availability: All data needed to evaluate the conclusions in the paper are present in the paper and supplementary materials. Source Data are provided in Data S1.”

“This work was supported by the Canadian Institutes of Health Research grant no. 400784 (to A. R.)”

5. We notice that your supplementary materials are included in the manuscript file. Please remove them and upload them with the file type 'Supporting Information'. Please ensure that each Supporting Information file has a legend listed in the manuscript after the references list.

Reviewers' comments:

Reviewer's Responses to Questions

**Comments to the Author**

1. Is the manuscript technically sound, and do the data support the conclusions?

Reviewer #1: Yes

Reviewer #2: Yes

2. Has the statistical analysis been performed appropriately and rigorously?

Reviewer #1: Yes

Reviewer #2: Yes

3. Have the authors made all data underlying the findings in their manuscript fully available?

Reviewer #1: Yes

Reviewer #2: Yes

4. Is the manuscript presented in an intelligible fashion and written in standard English?

Reviewer #1: Yes

Reviewer #2: Yes

Reviewer #1: Excellent manuscript, with some novel and interesting information presented. My comments are only minor, and refer mainly to the presentation of statistical output/detail.

In several of your figures, you present a series of plots. These are presented really well with individual points represented; however, in one or two cases there is a bar plot with error without points presented (or perhaps they are covered by error?). Please can these be presented as box-plots to allow the reader to gather further visual information about the variance between your replicates, while including the points in a similar way to the other plots.

In various places throughout the manuscript, a p-value is listed to support a finding, but the accompanying statistic and base information on data normality are missing. I cannot determine if data normality was checked, supporting statistic use - I don't believe this is specifically noted in the methods. Please can this be made clearer in the methods. In addition, where a p-value is listed, please include the statistic that was used to provide this value. This could also include the degrees of freedom and any accompanying value referring to variance (where relevant).

One point for the discussion. I wonder if you might elaborate on what your findings may mean for microsporidian culture. In many cases, lipid content in cell line choice is not a commonly explored parameter. Perhaps this is an important factor to consider for many species? If lipid levels change, this could have an effect on an ability to culture certain species.

Reviewer #2: The paper by El Jarkass and colleagues aimed to determine the links between lipidome and microsporidian infections in C. elegans.

The work has been very well performed, and represent an important addition to our current knowledge of microsporidian infections and how it affects the host metabolism.

I have no major comments, as I found the work to be well executed and timely. One suggestion would be to ensure that the results are placed into the context of the host species - for example, can we expect the lipidome of all or most hosts to be similarly affected, or does the C. elegans genome carry some uniqueness from that perspective?.

**Do you want your identity to be public for this peer review?** For information about this choice, including consent withdrawal, please see our Privacy Policy

Reviewer #1: **Yes: ** Dr Jamie Bojko

Reviewer #2: No

---

## [Author Response · Author response to Decision Letter 1]

11 May 2025

We thank the reviewers for their comments. We have provided a revised manuscript (where all changes are tracked) to address the reviewers’ concerns. Points raised by the reviewers have been incorporated into the manuscript and we have provided a point-by-point response to each of the reviewer’s comments:

Reviewer #1: Reviewer #1: Excellent manuscript, with some novel and interesting information presented. My comments are only minor, and refer mainly to the presentation of statistical output/detail.

We thank the reviewer for their kind comments.

In several of your figures, you present a series of plots. These are presented really well with individual points represented; however, in one or two cases there is a bar plot with error without points presented (or perhaps they are covered by error?). Please can these be presented as box-plots to allow the reader to gather further visual information about the variance between your replicates, while including the points in a similar way to the other plots.

Yes, there are several cases where we present normalized bar plots, where it is difficult to see the individual points. We have now changed these plots to just show the points and standard deviation. In once case, figure 2E, the control is normalized to 1, so there are no error bars to be shown in the control samples in this plot.

In various places throughout the manuscript, a p-value is listed to support a finding, but the accompanying statistic and base information on data normality are missing. I cannot determine if data normality was checked, supporting statistic use - I don't believe this is specifically noted in the methods. Please can this be made clearer in the methods. In addition, where a p-value is listed, please include the statistic that was used to provide this value. This could also include the degrees of freedom and any accompanying value referring to variance (where relevant).

To address this comment, we have added a statistics section to the methods and modified the legends to include the statistical test that was used for each figure panel. We also checked the normality of our data, and this information has been added to the statistics section of the methods.

One point for the discussion. I wonder if you might elaborate on what your findings may mean for microsporidian culture. In many cases, lipid content in cell line choice is not a commonly explored parameter. Perhaps this is an important factor to consider for many species? If lipid levels change, this could have an effect on an ability to culture certain species.

We have modified the discussion in lines 169 to 175 to mention how fatty acids can affect growth of other parasites in cell culture and microsporidia in several animals, and how this approach may be used to culture microsporidia such as E. bieneusi.

Reviewer #2: The paper by El Jarkass and colleagues aimed to determine the links between lipidome and microsporidian infections in C. elegans.

The work has been very well performed, and represent an important addition to our current knowledge of microsporidian infections and how it affects the host metabolism.

We thank the reviewer for their kind comments.

I have no major comments, as I found the work to be well executed and timely. One suggestion would be to ensure that the results are placed into the context of the host species - for example, can we expect the lipidome of all or most hosts to be similarly affected, or does the C. elegans genome carry some uniqueness from that perspective?

The strong impact of microsporidia on host lipid metabolism has now been reported in in fruit flies, shrimp, crabs, and silkworms and honey bees. Although there are differences in the exact lipid species being affected, lack of standardized experimental approaches make comparisons difficult and future work will be necessary to determine the importance of these differences. We have modified the discussion in lines 175 to 177 to mention and reference other hosts where changes to lipid metabolism caused by microsporidia have been observed.

---

## [Editor Report · Decision Letter 1]

Microsporidia infection alters C. elegans lipid levels

PONE-D-25-08621R1

Dear Dr. Reinke,

We’re pleased to inform you that your manuscript has been judged scientifically suitable for publication and will be formally accepted for publication once it meets all outstanding technical requirements.

Kind regards,

Paul Dean

Academic Editor

PLOS ONE

Additional Editor Comments (optional):

Thank you for your revised version of the manuscript and for addressing the points of the reviewers.
---

## [Editor Report · Acceptance letter]

PONE-D-25-08621R1

PLOS ONE

Dear Dr. Reinke,

I'm pleased to inform you that your manuscript has been deemed suitable for publication in PLOS ONE. Congratulations! Your manuscript is now being handed over to our production team.

Kind regards,

on behalf of

Dr. Paul Dean

Academic Editor

PLOS ONE